# Modeling of Stochastic Temperature and Heat Stress Directly Underneath Agrivoltaic Conditions with *Orthosiphon Stamineus* Crop Cultivation

**Noor Fadzlinda Othman [1,2], Mohammad Effendy Yaacob [2,3,4,*], Ahmad Suhaizi Mat Su [1], Juju Nakasha Jaafar [1], Hashim Hizam [4,5], Mohd Fairuz Shahidan [6], Ahmad Hakiim Jamaluddin [2], Guangnan Chen [7] and Adam Jalaludin [8]**

[1] Department of Agriculture Technology, Faculty of Agriculture, Universiti Putra Malaysia, Serdang, Selangor 43400, Malaysia; fadzlin013@gmail.com (N.F.O.); asuhaizi@upm.edu.my (A.S.M.S.); jujunakasha@upm.edu.my (J.N.J.)
[2] Hybrid Agrivoltaic Systems Showcase (HAVs) eDU-PARK, Universiti Putra Malaysia, Serdang, Selangor 43400, Malaysia; ahmadhakiimjamaluddin@gmail.com
[3] Department of Process & Food Engineering, Faculty of Engineering, Universiti Putra Malaysia, Serdang, Selangor 43400, Malaysia
[4] Centre for Advanced Lightning, Power and Energy Research (ALPER), Universiti Putra Malaysia, Serdang, Selangor 43400, Malaysia; hhizam@upm.edu.my
[5] Department of Electrical & Electronics Engineering, Faculty of Engineering, Universiti Putra Malaysia, Serdang, Selangor 43400, Malaysia
[6] Department of Landscape Architecture, Faculty of Design and Architecture, Universiti Putra Malaysia, Serdang, Selangor 43400, Malaysia; mohdfairuz@upm.edu.my
[7] Faculty of Health, Engineering and Sciences, University of Southern Queensland, Toowoomba, QLD 4350, Australia; Guangnan.Chen@usq.edu.au
[8] Department of Agriculture and Fisheries, Agri-Science Queensland, Leslie Research Facility, 13 Holberton Street, Toowoomba, QLD 4350, Australia; adam.jalaludin@gmail.com
\* Correspondence: fendyupm@gmail.com

**Abstract:** This paper presents the field measured data of the ambient temperature profile and the heat stress occurrences directly underneath ground-mounted solar photovoltaic (PV) arrays (monocrystalline-based), focusing on different temperature levels. A previous study has shown that a 1 °C increase in PV cell temperature results in a reduction of 0.5% in energy conversion efficiency; thus, the temperature factor is critical, especially to solar farm operators. The transpiration process also plays an important role in the cooling of green plants where, on average, it could dissipate a significant amount of the total solar energy absorbed by the leaves, making it a good natural cooling mechanism. It was found from this work that the PV system's bottom surface temperature was the main source of dissipated heat, as shown in the thermal images recorded at 5-min intervals at three sampling times. A statistical analysis further showed that the thermal correlation for the transpiration process and heat stress occurrences between the PV system's bottom surface and plant height will be an important factor for large scale plant cultivation in agrivoltaic farms.

**Keywords:** transpiration; PV heat conversion; plant heat stress; agrivoltaic system; sustainable integration; thermal analysis

## 1. Introduction

Dramatic changes and increasing public interest in solar photovoltaic (PV) landscapes show that the dual beneficial use of land may have better impacts on energy production and future agriculture

transdisciplinary design. Some highlights and recent research in solar PV projects by higher education institutions show that the solar industry has broadened its stakeholders and interest in the future, reflecting a significant shift in the dynamics of the market [1,2]. The PV industry for large scale solar projects is dominated by energy companies but, based on the effort above, it is shown that experts in higher education within the research environment have the capabilities to compete with energy companies in the solar PV industry. This trend has been transferred to ecological efficiency and positive effects, consequently upscaling the number and size of PV systems installed on the land. Rapidly decreasing price of PV modules in the world market in line with the increasing demand of fresh produce promotes the idea of agro-PV integration, commonly known as an agrivoltaic system.

This type of solar power system is a power generation system that incorporates several parts, namely PV modules, solar inverters, mounting, cabling and other electrical components, which are integrated in the balance of systems (BOS) [3,4]. This PV device absorbs rays from sunlight and translates them into a direct current (DC) via semiconductor materials. Malaysia, a tropical country in Southeast Asia, has given years of commitment to culturing green initiatives, especially PV systems and applications. This statement is evidenced by the increasing quota specifically for large scale solar (LSS) PV systems and the commitment by the Ministry of Energy, Science, Technology, Environment and Climate Change (MESTECC) [5] to persistently aim for a 20% energy mix by the year 2025 with multiple initiatives [6].

Generally, based on PV projects in University Putra Malaysia, where the size and ground conditions are put into a factor that generates empty areas under the panels, 1 kWp solar PV arrays may occupy roughly 8 to 12 square meters of land [7,8]. Based on their high demand, solar PV models in the market nowadays are ground-mounted arrays and require a fixed PV panel arrangement. There is a call for futuristic features from the market, with application in large-scale areas by enhancing their design while maintaining cost-effective deployment [9]. Temperature plays an important role in DC generation via PV modules. Park et al. [10], in their research on building-integrated PV (BIPV), defined such significant effects of the PV module's thermal characteristics, where approximately a 0.5% reduction in energy is generated based on a 1 °C increase of the module temperature. This statement is supported by Kim et al. [11], with additional information on the energy efficiency from a common PV module that can be increased due to a drop in surface temperature, especially on the highest heated portions of PV cells and ribbons.

The concept of agrivoltaics, or solar farming, aspired to creatively convert agriculture to photovoltaics, applied on the same land to maximize the yield [12]. The agrivoltaic system, as shown in Figure 1, contemplates specific plant attributes: height, productivity, water consumption and shading resistance. The figure demonstrates the idea of the agrivoltaic method employed in several countries by plotting vacant land with various types of crops. This method of farming under the solar panel is an innovation of incorporating green energy into agriculture and it is a part of introducing modern aspects to the agricultural community [13]. Some of the published results in [9,12–14] relating to agrivoltaic projects summarized the importance and successful integration of the systems by assessing whether:

- The AV system improved environmental efficiency.
- The AV system promoted effective usage of light and space for concurrent energy and food output.
- The AV system boosted the technological capacity for PV and agricultural production conjointly by implementing a hybrid simulation model.
- The AV system yielded more crop as compared to the period before the deployment.

This integrated system will maximize crop production, enhancing the system's performance while addressing land management and sustainability issues. The integration of these two resources would optimize the yield, improve clean system efficiency and solve the issue of land resource sustainability. The issue of the agrivoltaic concept implemented in ground-mounted PV systems and the shading effect of the PV arrays on crop canopy have been discussed by [15] recently.

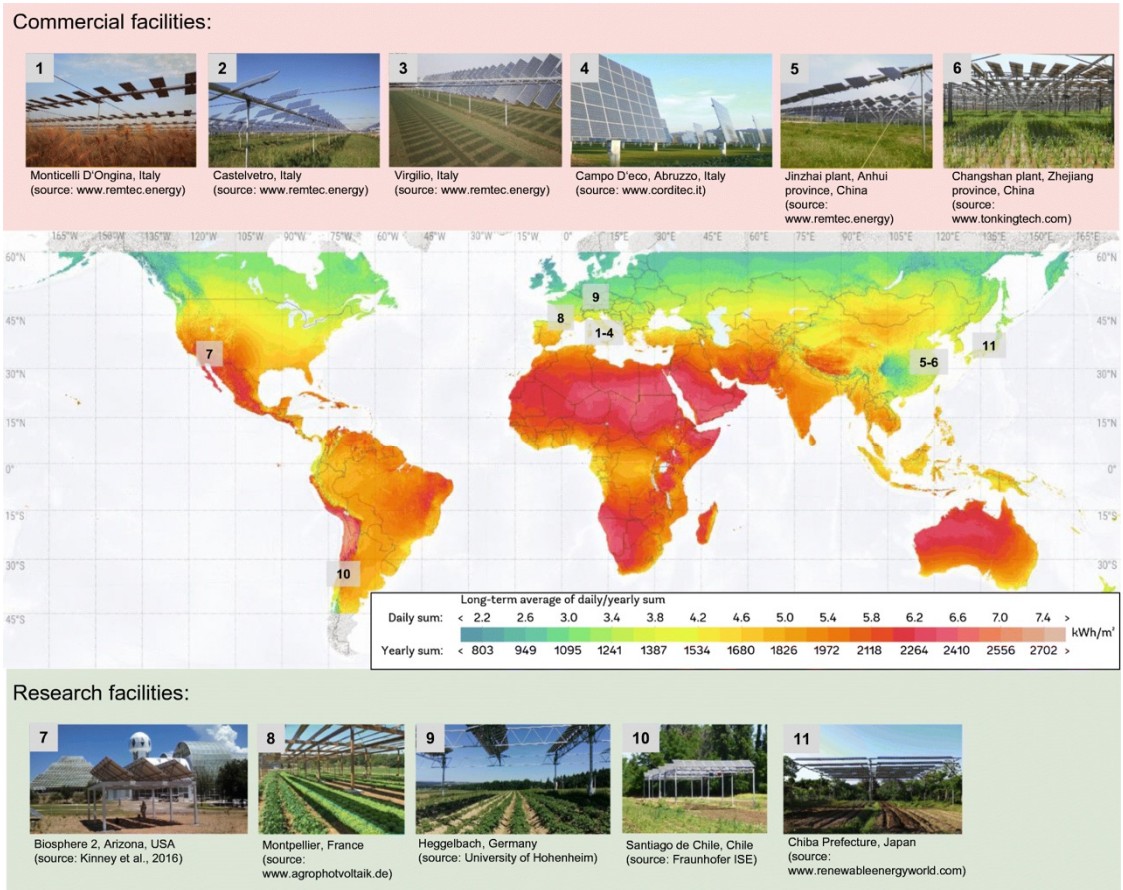

**Figure 1.** Typical agrivoltaic research facilities worldwide. [15].

The group suggested that the density of the PV arrays should be reduced adequately to enable ample amounts of light penetration while also maintaining a respectable production of DC electricity. The concept of agrivoltaics is in line with the Kyoto Protocol [16] and the United Nations Sustainable Development Goals (UN-SDG) [17,18], which promote the usage of clean and affordable energy towards sustainable urban infrastructure and further reducing the usage of fossil fuels.

In Malaysia, most planned and retrofitted agrivoltaic facilities are based on existing ground-mounted solar PV farm infrastructures where the primary activity is to sell the electricity generated to the National Grid. The issue of ground-mounted photovoltaic systems can be explained based on several factors, namely:

- The fact that existing solar PV farms do not allow any intervention or disturbance to any wiring, operation, structure or subsurface of the PV systems.
- The difficulties and hazards for farmers working under PV arrays result in lower production yield.
- Semi-confined working spaces, as workers have to bend down and inspect plants under PV array structures for growth monitoring and harvesting activities.
- The need for some tools to ease the process of planting, harvesting and post-harvest under agrivoltaic farming (most crop yields four cycle harvest per annum).

Heat stress normally occurs when temperatures rise above a certain level for a certain period and bear deleterious and permanent effects on a crop cycle, thus affecting yield [19,20]. Generally, heat stress is set to occur when a transient temperature rises over the average temperature of 10–15 °C [20–25]. The degree to which it happens in a particular climate zone relies on the frequency and amount of extreme temperatures happening during the day and/or the night. Some general definitions by [20] have also discussed the tendency of plants to grow with good economic yield under high temperature

conditions. The extent to which this occurs in specific climatic zones depends on the probability and period of high temperatures occurring during the day and/or the night.

The transpiration process plays an important role in the cooling of green plants where, on average, it could dissipate around 32.9% of the total solar energy absorbed by the leaves, making it a good natural cooling mechanism [26–28]. However, the magnitude of its impact varies from species to species. Increased transpiration levels do have an impact on water stress because the increase in ambient temperature increases the water evaporation from ground soil, thus, some plants have a tendency to grow slowly or even die at an early stage. *Orthosiphon stamineus* was chosen as the herbal plant for a project where, based on field evaluation (40 days under tropical climate), remarkably, the crop proved growth sustainability [29]. Compared to the four other types of herbal plants in the assessment, *Orthosiphon stamineus* showed healthy growth and its morphological aspects were enhanced compared to the normal conditions. The roots and fresh branches showed aggressive growth, mostly due to the soil's moisture content, thus, it could be harvested on time. The method of cultivation underneath solar PV arrays used a drip fertigation system (DFS) directly to polybags, to maintain the soil's moisture level and to prevent any disturbances to the electrical cablings and trenches. This method also eased the process of harvesting and replanting under such restricted conditions.

Herbal plants tend to possess valuable bioactive chemical compound reserves with an abundance of possible applications in pharmaceutical and agrochemical industries. [30] explained the basic concept of microclimate conditions as a set of climate parameters assessed in a specified area near the surface of the planet, including a variation of temperature, light, wind intensity and relative humidity (RH), which are significant measures for habitat selection and other ecological practices. One of the critical elements calculated based on these parameters was the vapor pressure deficit (VPD), which is defined as the discrepancy between the volume of moisture in normal settings with saturated condition (VPD in a greenhouse range of 0.45 kPa to 1.25 kPa with an idle of 0.84 kPa) [30]. Leonardi, Guichard and Berlin, in [31], explained that during daylight hours, where the high VPD condition was enhanced, the transpiration rates were better for plants to grow because the VPD exerted a substantial rise of soluble solids but lowered the fruits' fresh weight and internal fluid levels. A plant's transpiration, and the correct VPD under a controlled environment, can effectively help to optimize the plant's ideal growth and plant health [32,33]. Hot and dry surrounding air under shade can produce high VPD and causes stress to the plant.

In agrivoltaic systems, plants, or crops, are one of the crucial elements that need to be considered. The transpiration process in plant growth takes place when water is biologically released from the aerial parts of the plants in the form of water vapor. During the process of transpiration, as illustrated in Figure 2, water molecules are transmitted from roots to stomata, the small pores underneath the leaves, where vaporization takes place, and the molecules are transpired through the surrounding air. The effect of vaporization increases with the number of plants being deposited under the PV panels, which results in an increased RH value.

Crawford et al., in [28], explained that extreme temperatures multiply the risk of plant damage due to the heat and, simultaneously, water shortage, which enhances the plant cooling capability, as shown in Figure 3. The increase in transpiration rate is directly correlated with the increased in stomata opening thus, this increases photosynthesis activities.

The transpiration characteristics of plants in different surrounding temperatures and relative humidities portray a significant heat dissipation value (transpirative heat transfer through leaves). In relation to this, a study by [27] in Wuxi, China, during the summer and winter seasons reflected a 55.8% and 24.3% transpiratory heat flux for each season, respectively, accounting for the total heat dissipation of the cinnamon. Temperature difference, $\Delta T$, is a crucial factor to be analyzed in agrivoltaic conditions, especially the effect of plant height for each growth cycle. Mittler, in [35], explained that heat is one of the prominent elements in the abiotic stress effect on plant growth where, during heat stress, plants open their stomata to cool their leaves by transpiration. If the condition is prolonged or under an increasing rate, this will eventually create a greater detrimental effect on the plant's

growth and productivity. Therefore, this study aims to measure the ambient temperature profile and the impacts of heat stress occurrences directly underneath ground-mounted solar PV arrays, focusing on different temperature levels.

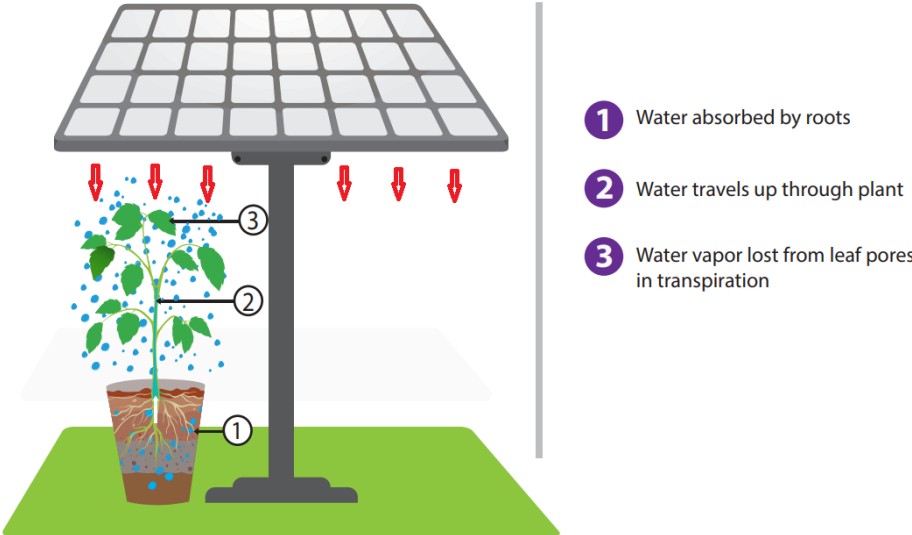

**Figure 2.** A simple analogy of the plant transpiration process directly underneath a photovoltaic (PV) module (heat source). Original source from [34].

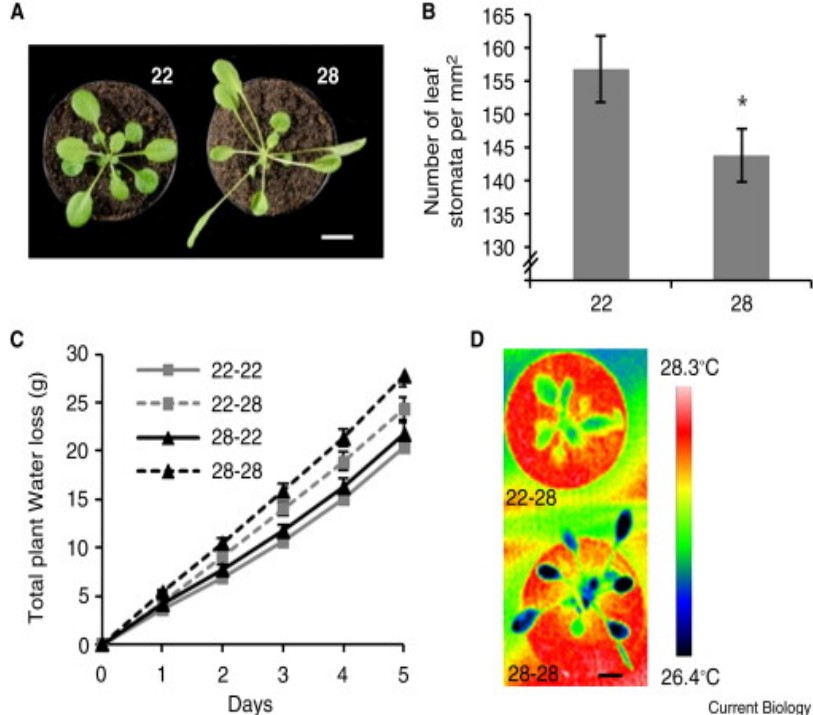

**Figure 3.** Crop responses at high temperatures indicate an increase in transpiration and enhanced leaf cooling capacity. (**A**) shows the plant at two different temperature levels and the thermal image of this condition is shown in (**D**), (**B**) proves the increasing number of leaves at lower temperature with respect to the lower value of water loss as shown in (**C**).

## 2. Methodology

This work was carried out based on a straightforward process so as to study the actual effects of temperature on planting cultivations under agrivoltaic conditions, comprising site setup, installation of sensors, data loggers, weather stations and thermal imagers, with an emphasis on the statistical analysis of the field temperature parameters.

### 2.1. Site Setup

The site setup was located at the Hybrid Agrivoltaic System Showcase (HAVs), Faculty of Engineering, University Putra Malaysia. A weather station was installed on site to measure the environmental factors. The location of the station was near the PV array at a 2 m height to negate any ground disturbances, whilst the PV structure height ranged from 1 m to 1.5 m. The Arduino-based data acquisition (DAQ) compartment, type-K thermo sensor and wind sensor are shown in Figure 4a.

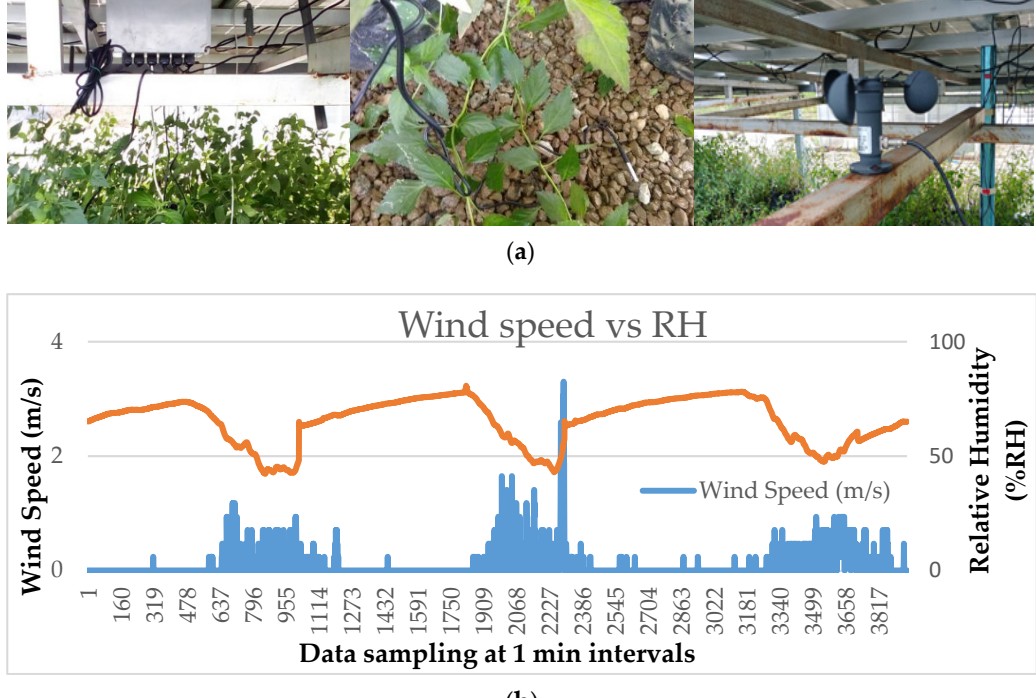

(**a**)

(**b**)

**Figure 4.** (**a**) Installation of the data acquisition (DAQ) compartment, thermo sensor and other environmental sensors; (**b**) Data plots for relative humidity (RH) and wind speed for agrivoltaic plots.

Based on 24 h data monitoring, shown in Figure 4b, a total of 3956 data samples were recorded for temperature value (°C), wind speed (m/s) and RH. It was observed that the average wind speed was only 0.098 m/s, due to the stagnant condition most of the time and the location of the wind sensor under the PV array (approx. 4 feet from ground level). The maximum recorded wind speed was 3.3 m/s. The maximum value for RH was 80.71%, with an average reading of 65.67% throughout the three-day duration.

The ambient temperature surrounding the plant leaves was the main component to be recorded and analyzed in this project. A Fluke thermal imager was used to record videos and images of surrounding temperatures and it was located at a 2 feet distance from the edge of the PV array, with an infrared lens focusing on the leaves (middle angle), as shown in Figure 5.

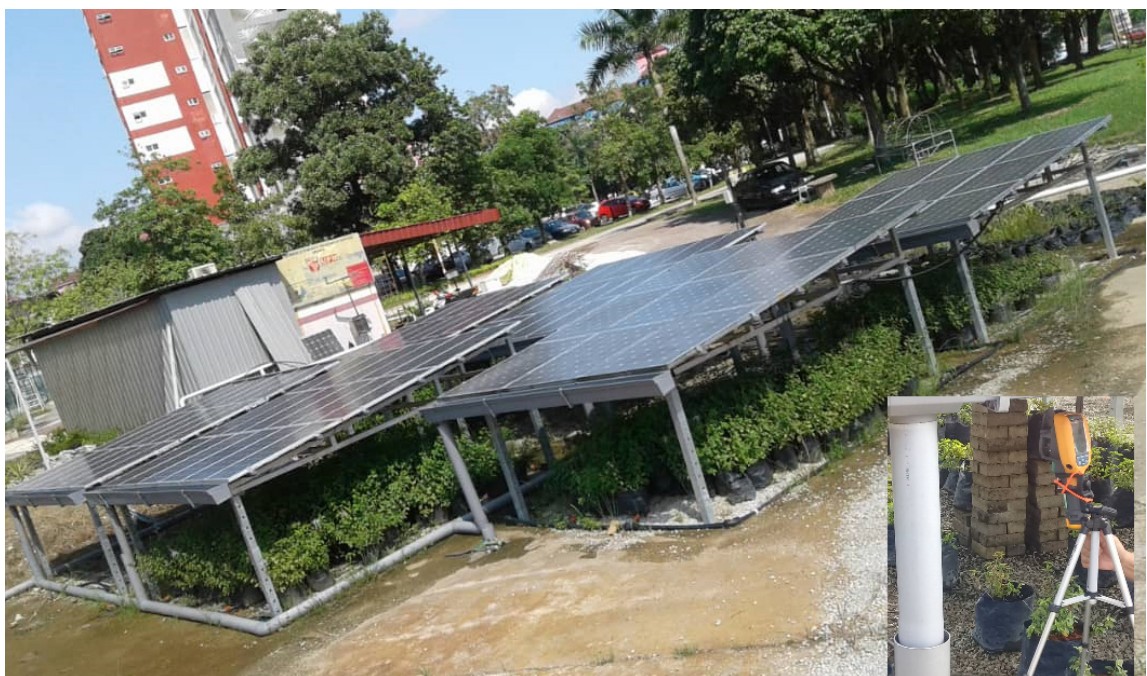

**Figure 5.** Agrivoltaic system with a Fluke thermal imager on a tripod for video recording.

### 2.2. Calculation for Vapor Pressure Density

Vapor pressure density (VPD) in kilopascals (kPa) can be measured by subtracting the actual vapor pressure of the air with the saturated vapor pressure ($VP_{sat} - VP_{air}$), as shown in Equation (1).

$$VPD = VP_{sat} - VP_{air} \tag{1}$$

where:

$VP_{sat} = T_a/1000$
$VP_{air} = VP_{sat} \times RH/100$

The value for VPD was also summarized and simplified by the University of Arizona's College of Agriculture and Life Sciences [36] using their online VPD calculator, where the user only inserts the values for air temperature ($T_a$) and relative humidity. The information related to the microclimate for a specified location reflects the ecological processes and wildlife behavior, covering some elements of plant regeneration and growth which depict their unique spatial and temporal responses to change [37,38]. It is also a crucial measure to identify permutations in the local environment for tracking and evaluating the results of various management regimes.

Extreme high-temperature events affect the demand for atmospheric water vapor, which could be represented by the energy balance of a leaf, shown in Equation (2).

$$S_t(1 - a_\iota) + L_d - \varepsilon\sigma T_\iota^4 = \frac{pC_p\,(T_\iota - T_a)}{r_a} + \frac{pC_p(e^* - e_a)}{r(r_s - r_a)} \tag{2}$$

where:

$S_t$ is the incoming solar radiation,
$a_\iota$ is the albedo of the leaf or canopy,
$L_d$ is the incoming longwave radiation,
$\varepsilon$ is the emissivity of the leaf or canopy,
$\sigma$ is the Stefan–Boltzmann constant,

$T_t$ is the leaf canopy temperature,
$T_a$ is the ambient temperature,
$p$ is the density of dry air,
$C_p$ is the volumetric heat capacity of dry air,
$r_a$ is the aerodynamic conductance,
$r_s$ is the canopy conductance,
$e^*$ is the saturation vapor pressure,
$e_a$ is the saturation ambient pressure.

Saturation vapor pressure ($e^*$) is exponentially relative to air temperature, thus, the changing of the $e^*$ value would affect the energy balance. Based on this correlation, an increase in VPD causes more water to be transpired by a leaf, leading to a reduction in photosynthesis [39].

Thermal images using the Fluke device are shown in Figure 6, where all the thermal images were taken using the same device and the same PV panel arrangements at different times of shooting (Figure 6 shows the thermal conditions at 11 a.m. and 3 p.m.). The images show a much higher temperature below the PV panels, which was reflected in the surrounding temperature condition and in the scope directly underneath the PV panels. A sample video clip of the thermal conditions underneath the PV array is enclosed with the document.

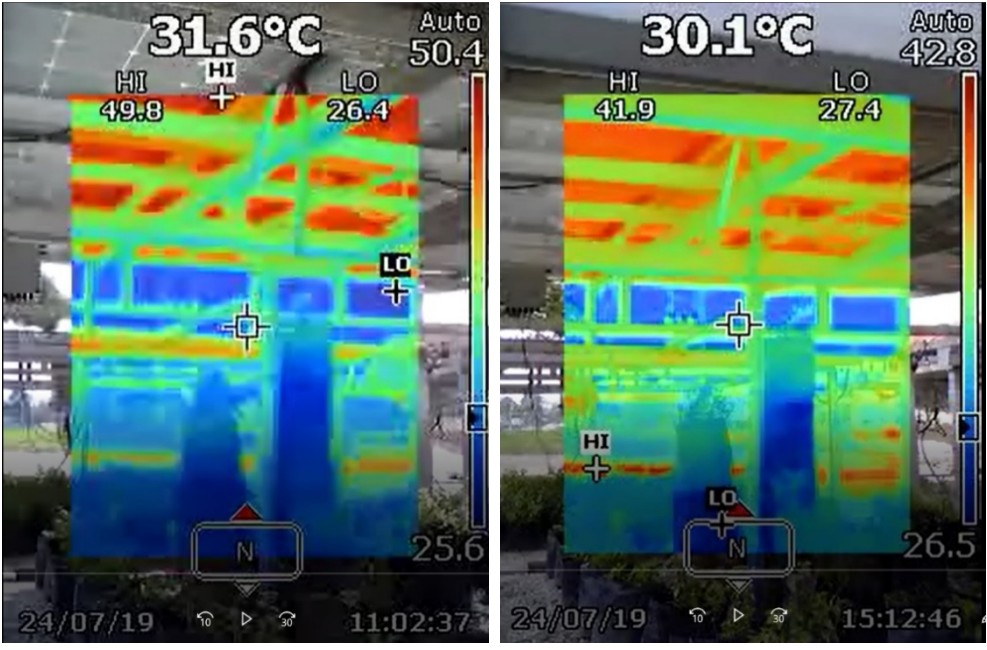

**Figure 6.** Thermal images of the agrivoltaic conditions for hourly sampling at the University Putra Malaysia (UPM) site.

The thermal imager provided some insight into the temperature under agrivoltaic conditions, although the readings might not be too precise because they only showed one spot value at a time. Figure 6 and Video S1 show the temperature values at different locations, i.e., below the PV panel, the surrounding air underneath PV, the surrounding air at plant level, around the leaves and the ground surface temperature taken randomly at different times (5-min intervals). Assumptions were made for the temperature values at each location and level based on the color indicator on the right side.

## 3. Results and Discussion

The contribution from this work can be shown in the temperature elements plotted in Figure 7, where the actual temperature pattern for six different heights under agrivoltaic conditions is portrayed,

using 3600 data samples for five consecutive days from 7 a.m. to 7 p.m., daily. Each temperature value came from a thermal sensor (Type K: DS18B20, Maxim Integrated, San Jose, CA, US), starting from $T_g$, which was the ground surface temperature, up to the bottom of the PV array, ($T_{b,pv}$) which was directly glued to the PV array's bottom surface. The other four temperature locations ($T_{1ft,2ft,3ft,4ft}$) were based on readings from a hanging sensor to measure the surrounding air temperature.

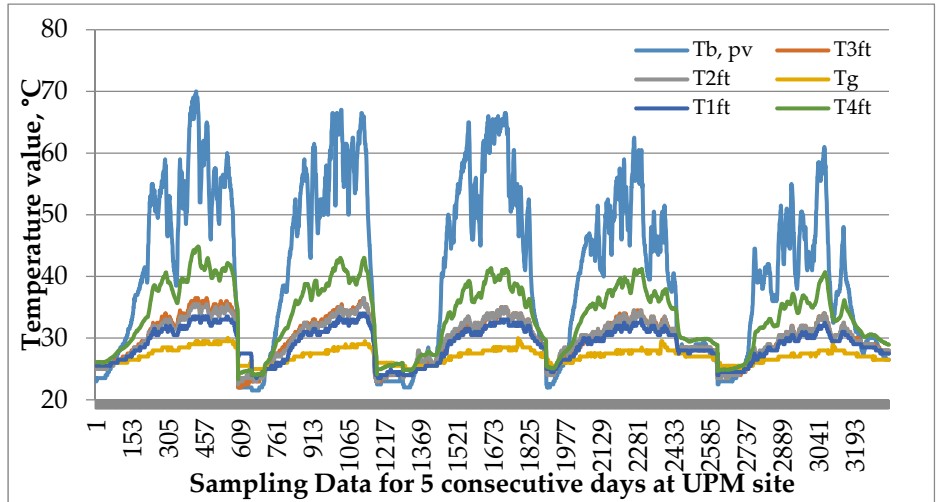

**Figure 7.** Temperature trends under agrivoltaic conditions at 1 min intervals (12 h daily). Abbreviations: $T_g$: Ground temperature; $T_{1ft,2ft,3ft,4ft}$: Temperature at 1 foot intervals; $T_{b,pv}$: PV panel's bottom surface temperature.

Based on the temperature values in Table 1, the maximum recorded temperature for $T_{1ft}$, $T_{2ft}$ and $T_{3ft}$ was 34 °C, 36.5 °C and 36.5 °C, respectively, where, at this height, the plant started growing under agrivoltaic conditions. The value for $\Delta T_{max}$ was increasing with the plant height–temperature difference (1–2 feet) ranging below 3 °C. The ground temperature ($T_g$) was considered as the reference value based on its effect on plant seedlings, and $T_b$ (the bottom surface of PV module) as the maximum plant height. Hatfield and Prueger [39] explained that the rate of plant growth and development is heavily dependent on the surrounding temperature (min, max and optimum temperature values) and the annual temperature increment due to global warming over the next 50 years is likely to reach 1.5 °C between 2030 and 2052 [40].

**Table 1.** Values of temperature difference, ΔT, (in °C) based on a 1 foot height distribution.

|  | $T_g$ | $T_{1ft}$ | $T_{2ft}$ | $T_{3ft}$ | $T_{4ft}$ | $T_{b,pv}$ |
|---|---|---|---|---|---|---|
| Average | 27.14 | 29.06 | 29.78 | 29.83 | 33.47 | 40.97 |
| Max | 30 | 34 | 36.5 | 36.5 | 44.88 | 70 |
| Min | 25 | 23.5 | 22.5 | 22 | 23.33 | 21.5 |
| $\Delta T_{ave}$ | 1.92 | 2.64 | 2.69 | 6.33 | 13.83 | |
| $\Delta T_{max}$ | 4 | 6.5 | 6.5 | 14.88 | 40 | |
| $\Delta T_{min}$ | −1.5 | −2.5 | −3 | −1.67 | −3.5 | |

Abbreviations: $T_g$: Ground temperature; $T_{1ft,2ft,3ft,4ft}$: Temperature at 1 foot intervals; $T_{b,pv}$: PV panel's bottom surface temperature.

Based on Equation (1) and an online calculator software, the values for VPD are summarized in Table 2. The value for $T_{1ft}$ was used to represent the designated surrounding air temperature ($T_a$) because the location was at par with the plant at a 1 foot height and touching the polybags and soil.

**Table 2.** Vapor pressure density (VPD) calculations based on 1 foot height under agrivoltaic conditions.

| Reading | Ta (°C) | % RH | SVP (kPa) | VP (kPa) | VPD (kPa) |
|---|---|---|---|---|---|
| Average Value | 27.24 | 70.36 | 3.618 | 2.546 | 1.072 |
| Max Value | 34 | 89.7 | 5.324 | 4.776 | 2.005 |
| Min Value | 23.5 | 30.77 | 2.897 | 0.891 | 0.548 |

Abbreviations: $T_a$: Ambient temperature; % RH: Relative humidity; SVP: Saturated vapor pressure; VP: Vapor pressure; VPD: Vapor pressure density.

The optimum value for VPD under a greenhouse condition ranges from 0.45 kPa to 1.25 kPa, ideally sitting at around 0.85 kPa [31]. For agrivoltaic conditions, the VPD value ranged between 2.005 kPa (max) to 0.548 kPa (min), with an average value of 1.072 kPa.

For the temperature analysis, the field data measured were segregated into five sampling hours (daily) with different temperature levels, as shown in Table 3.

**Table 3.** Analysis of temperature distributions based on sampling hours.

| | | Measure | Early Sun | Moderate Sun (Morning) | Peak Sun | Moderate Sun (Afternoon) | Mild Sun (Evening) |
|---|---|---|---|---|---|---|---|
| **Time** | | | 7:00–8:59 | 9:00–10:59 | 11:00–14:59 | 15:00–16:59 | 17:00–18:59 |
| **Temperature** | Tg (°C) | Average | 25.5409 | 26.5125 | 27.9233 | 28.2828 | 26.3924 |
| | | Min | 25.0000 | 25.5000 | 26.5000 | 25.5000 | 25.5000 |
| | | Max | 26.5000 | 27.5000 | 29.5000 | 30.0000 | 27.5000 |
| | T1ft (°C) | Average | 25.1210 | 27.8650 | 31.4333 | 30.7819 | 26.9340 |
| | | Min | 23.5000 | 25.0000 | 28.0000 | 24.0000 | 23.5000 |
| | | Max | 27.5000 | 30.5000 | 33.5000 | 34.0000 | 29.0000 |
| | T2ft (°C) | Average | 25.2473 | 28.7383 | 32.6763 | 31.6755 | 26.4902 |
| | | Min | 23.0000 | 25.0000 | 28.5000 | 23.0000 | 22.5000 |
| | | Max | 29.5000 | 31.5000 | 35.5000 | 36.5000 | 29.5000 |
| | T3ft (°C) | Average | 25.0249 | 28.6808 | 32.8779 | 31.8989 | 26.3888 |
| | | Min | 23.0000 | 25.0000 | 28.5000 | 23.0000 | 22.0000 |
| | | Max | 28.0000 | 31.5000 | 36.5000 | 36.5000 | 29.5000 |
| | T4ft (°C) | Average | 26.0758 | 32.0005 | 38.4923 | 35.7665 | 27.9299 |
| | | Min | 24.0000 | 26.1000 | 31.9500 | 26.9100 | 23.3300 |
| | | Max | 30.1600 | 37.6400 | 44.8800 | 43.0300 | 31.1700 |
| | Tb, pv (°C) | Average | 24.6806 | 39.9392 | 53.6071 | 42.3635 | 26.0416 |
| | | Min | 21.5000 | 26.0000 | 35.0000 | 23.0000 | 22.0000 |
| | | Max | 35.5000 | 55.0000 | 70.0000 | 66.5000 | 30.5000 |

Abbreviations: $T_g$: Ground temperature; $T_{1ft,2ft,3ft,4ft}$: Temperature at 1 foot intervals; $T_{b, pv}$: PV panel's bottom surface temperature.

Based on Table 3 and R programming, the heat stress contour throughout the five sampling hours was plotted as shown in Figure 8.

An illustration of heat stress occurrences in % value with respect to the 1 foot height–temperature level under agrivoltaic conditions is shown in Figure 8. These field data were further analyzed as shown in Figure 9, where dependencies on the bottom of the PV panel and at a 4 foot height can be observed.

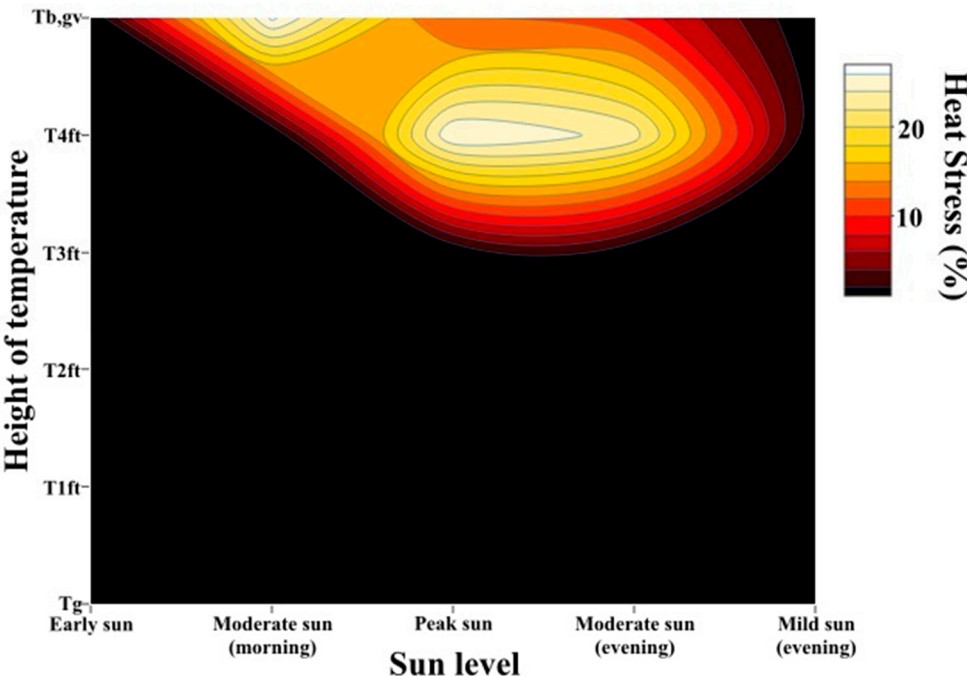

**Figure 8.** Heat stress occurrences (%) at five sampling hours.

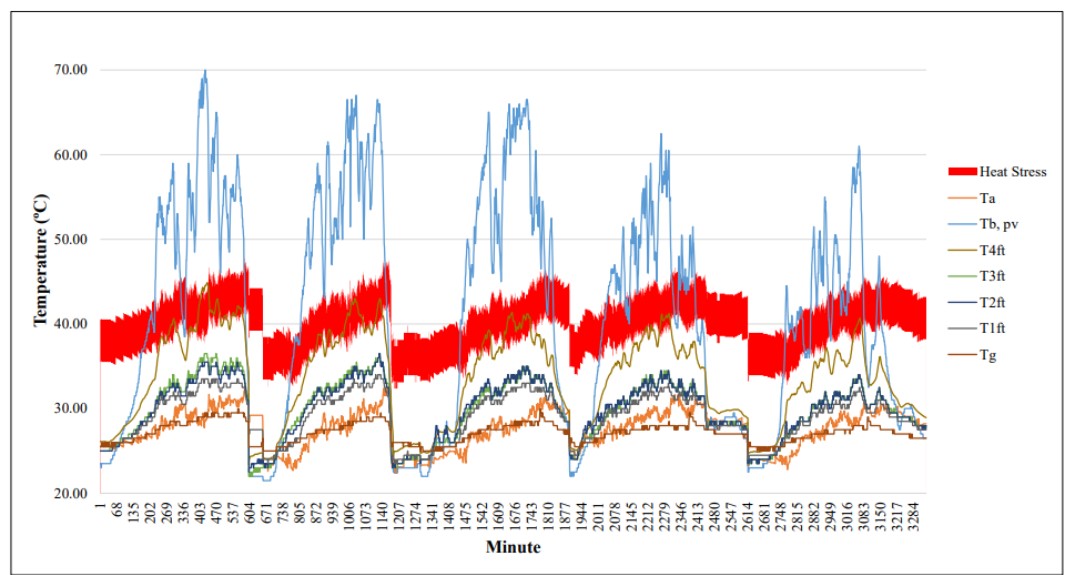

**Figure 9.** Field observation for heat stress directly underneath the PV arrays. Abbreviations: $T_a$: Ambient temperature; $T_g$: Ground temperature; $T_{1ft,2ft,3ft,4ft}$: Temperature at 1 foot intervals; $T_{b, pv}$: PV panel's bottom surface temperature.

Based on Figure 8, the percentage of heat stress occurrences shows at what specific time in the day the plant will possibly experience a high surrounding temperature, above the normal ambient temperature. Based on the data sample, the highest heat stress occurred at a 4 foot height during peak sun and moderate sun (afternoon), with more than 23% heat stress points, as shown in Table 4. This is due to the bottom of the PV panel producing a much higher temperature after the photonic conversion and heat dissipation process. The ground heat's effect in this agrivoltaic condition was relatively low due to the PV array shading, as per temperature values for $T_g$ until $T_{2ft}$, thus, it can be assumed that no heat stress was caused by this.

**Table 4.** Percentages of heat stress ($T_h$) occurrence across sun level and height.

|  | Temperature Level | Early Sun | Moderate Sun (Morning) | Peak Sun | Moderate Sun (Afternoon) | Mild Sun (Evening) |
|---|---|---|---|---|---|---|
| **Time** |  | 7:00–8:59 | 9:00–10:59 | 11:00–14:59 | 15:00–16:59 | 17:00–18:59 |
| **Percentage of Occurrence (%)** | Tg | 0 | 0 | 0 | 0 | 0 |
|  | T1ft | 0 | 0 | 0 | 0 | 0 |
|  | T2ft | 0 | 0 | 0 | 0 | 0 |
|  | T3ft | 0 | 0 | 0 | 0 | 0 |
|  | T4ft | 0 | 0 | 25.9167 | 23.2270 | 0 |
|  | Tb, pv | 0 | 26.5000 | 10.3333 | 9.3972 | 0 |

A two-sample proportion test and a Chi-square test were used as the statistical approaches as shown in Tables 5 and 6, respectively.

**Table 5.** Count of heat stress ($T_h$) cases across temperature–height levels during peak sun.

|  |  | Heat Stress Status | | Total |
|---|---|---|---|---|
|  |  | Heat Stress | Non Heat Stress |  |
| **Temperature–Height and Sun Level** | $T_{4ft-PeakSun}$ | 311 | 889 | 1200 |
|  | $T_{bpv\_PeakSun}$ | 124 | 1076 | 1200 |

**Table 6.** Chi-square test for difference in proportions of heat stress ($T_h$) occurrence during peak sun.

|  | Value | df | Asymptotic Significance (2-Sided) | Asymptotic Significance (1-Sided) |
|---|---|---|---|---|
| **Pearson Chi-Square** | 98.184 | 1 | 0.000 | 0.000 |

Based on the Chi-square test, $T_{4ft}$ had a higher percentage of heat stress occurrence than $T_{b,pv}$ during peak sun at 99% confidence level ($p < 0.00001$). The same test was conducted for height level during moderate sun (afternoon) and these results also proved that $T_{4ft}$ had a higher percentage of heat stress occurrence than $T_{b,pv}$ during moderate sun (afternoon) at 99% confidence level ($p < 0.00001$).

Based on the correlations of $T_{b,pv}$ and $T_{4ft}$ towards heat stress ($T_h$) under agrivoltaic conditions, a summary of the findings of both the minimum and maximum values of heat stress, $T_{h,min}$ and $T_{h,max}$, is modelled as shown in Table 7. Some preliminary assessments were conducted to assess the fitness of data for regression modelling and the findings are displayed in Figures S1a–d, S2a–d, S3a–b, S4a–b and Table S7. Since all assumptions were fulfilled, regression models were developed and detailed findings are presented in Table S1–S6 which were simplified into Tables 7 and 8. The coefficient of determination (R squared) was 0.739, which indicates that 73.9% of the variation in $T_{h,min}$ and $T_{h,max}$ could be explained by the variation in both $T_{b,pv}$ and $T_{4ft}$, and both the $T_{h,min}$ and $T_{h,max}$ models were significantly fit at a 99% confidence level ($F = 4724.462$, $p$-value $< 0.001$).

**Table 7.** Regression statistics and analysis of variance (ANOVA).

|  | *df* | *SS* | *MS* | *F* | *Significance F* |
|---|---|---|---|---|---|
| **Regression** | 2 | 13,722.334 | 6861.167 | 4724.462 | 0.000 |
| **Residual** | 3332 | 4838.944 | 1.452 |  |  |
| **Total** | 3334 | 18,561.278 |  |  |  |

Multiple R = 0.860; R Square = 0.739; Adjusted R Square = 0.739; Standard Error = 1.205; Observation Counts = 3335.

**Table 8.** Individual t-test on independent variables.

|  | Coefficients | Standard Error | t Stat | *p*-Value | Lower 95% | Upper 95% |
|---|---|---|---|---|---|---|
| $T_{h,max}$ **Intercept** | 21.553 | 0.232 | 93.023 | 0.000 | 21.098 | 22.007 |
| $T_{h,min}$ **Intercept** | 16.553 | 0.232 | 71.442 | 0.000 | 16.098 | 17.007 |
| $T_{b,pv}$ | −0.293 | 0.005 | −57.141 | 0.000 | −0.303 | −0.283 |
| $T_{4ft}$ | 0.987 | 0.013 | 78.155 | 0.000 | 0.962 | 1.011 |

A t-test on independent variables, as shown in Table 8, confirmed that both $T_{b,pv}$ and $T_{4ft}$ significantly affected the $T_{h,min}$ and $T_{h,max}$ at 99% confidence level ($t_{Tb,pv}$ = −57.141, $t_{T4ft}$ = 78.155; *p*-value < 0.001). Hence, both were significant predictors of $T_{h,min}$ and $T_{h,max}$. Meanwhile, a unit increase of $T_{b,pv}$, $T_{h,min}$ and $T_{h,max}$ would decrease by 0.293 °C, and a unit increase in $T_{4ft}$ would increase $T_{h,min}$ and $T_{h,max}$ by 0.987 °C.

$T_{h,min}$ and $T_{h,max}$ could be expressed by the following new equations:

$$T_{h,min} = 16.553 - 0.293T_{b,pv} + 0.987T_{4ft} \tag{3}$$

$$T_{h,max} = 21.553 - 0.293T_{b,pv} + 0.987T_{4ft} \tag{4}$$

Or both equations could be simplified into a heat stress temperature model:

$$T_h \text{ (Heat stress temperature)} = [16.553, 21.553] - 0.293T_{b,pv} + 0.987T_{4ft} \tag{5}$$

## 4. Conclusions

As a major source of renewable energy, many photovoltaic farms have now been constructed in the world. The agrivoltaic system is a further concept that aims to combine commercial agriculture and photovoltaic electricity generation in the same space, in order to maximize crop production while addressing land management and sustainability issues.

This paper has presented the field measured data of ambient temperature profile and the heat stress occurring directly underneath solar photovoltaic (PV) arrays (monocrystalline-based) in a tropical climate condition (in Malaysia). With reference to the plant heat stress at 10 °C to 15 °C above the ambient temperature, the percentage of heat stress occurrences was the highest at a 4 foot height during peak sun and moderate sun (afternoon), with more than 23% heat stress points. It has also been found that the ground heat effect in this agrivoltaic condition was relatively low due to the PV array shading. A heat stress model for ground-mounted agrivoltaic conditions has been developed. It has been found that the coefficient of determination (R squared) for the model is 0.739, indicating that 73.9% of variation in $T_{h,min}$ and $T_{h,max}$ could be explained by the variations in both $T_b$, pv and $T_{4ft}$. Both $T_{h,min}$ and $T_{h,max}$ models were significantly fit at 99% confidence level. This paper has contributed to the understanding of plant physiological processes in response to environmental conversion factors. The model developed could also be used for further exploring the integration of crop cultivation and PV energy generation for optimum land use.

**Supplementary Materials:** The following are available online at http://www.mdpi.com/2073-4395/10/10/1472/s1, Table S1: Regression statistics of $T_{h,min}$ Model, Table S2: Analysis of variance (ANOVA) $T_{h,min}$ Model, Table S3: Individual t-test on independent variable $T_{h,min}$ Model, Table S4: Regression statistics $T_{h,max}$ Model, Table S5: Analysis of variance (ANOVA) $T_{h,max}$ Model, Table S6: Individual t-test on independent variable $T_{h,max}$ Model, Table S7: Variance inflation factor (VIF) for all independent variables, Figure S1: Boxplots for outlier detection. (a) Tb,pv; (b) T4ft; (c) Th,min; (d) Th,max, Figure S2: Scatter plots between dependent and independent variables for linearity. (a) Tb,pv against Th,min; (b) T4ft against Th,min; (c) Tb,pv against Th,max; (d) T4ft against Th,max, Figure S3: Normal QQ plot for residuals for normality; (a) Th,min model; (b) Th,max model, Figure S4: Residuals against fitted values plots for homoscedasticity; (a) Th,min model; (b) Th,max model.

**Author Contributions:** Conceptualization, N.F.O., M.E.Y., and A.S.M.S.; Methodology, N.F.O.; Software, A.H.J., N.F.O.; Validation, formal analysis, and investigation, N.F.O., M.E.Y., A.S.M.S., and A.H.J.; Resources, M.E.Y. and A.S.M.S.; Data curation, N.F.O. and A.H.J.; Writing—original draft preparation, N.F.O. and M.E.Y.; Writing—review and editing, A.S.M.S., J.N.J., H.H., M.F.S., G.C., and A.J.; Visualization, N.F.O.; Supervision,

M.E.Y. and A.S.M.S.; Project administration, N.F.O.; Funding acquisition, M.E.Y. and A.S.M.S. All authors have read and agreed to the published version of the manuscript.

**Funding:** This research was funded by the Ministry of Energy, Science, Technology, Environment and Climate Change (MESTECC) under the MESITA (Malaysia Energy Supply Industry Trust Account) Research Fund (Vote no. 6300921), and the Research Management Center (RMC), University Putra Malaysia, for the approval of research funding under the IPS Putra Grants Scheme (Vote no. 9667400).

**Acknowledgments:** The authors delegate our thanks to the Ministry of Energy, Science, Technology, Environment and Climate Change (MESTECC) under the MESITA (Malaysia Energy Supply Industry Trust Account) Research Fund (Vote no. 6300921) and the Research Management Center (RMC), University Putra Malaysia, for the approval of research funding under the IPS Putra Grants Scheme (Vote no. 9667400).

**Conflicts of Interest:** The authors declare no conflict of interest.

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
