# Peer review of "Modeling of Stochastic Temperature and Heat Stress Directly Underneath Agrivoltaic Conditions with Orthosiphon Stamineus Crop Cultivation"

_agronomy, doi:10.3390/agronomy10101472_

Round 1
Reviewer 1 Report
The revised manuscript of "modeling of stochastic temperature and heat stress directly underneath agrivoltaic conditions with orthosiphon stamineus crops cultivation" by Othman an co-authors gives extra information that help consolidating the methodological choices and the frame of the study. I have 2 comments on the revised manuscript.
1) One underlying reason of the study is the cooling from the panel side, but is not treated in this manuscript, as it is focused on the heat stress caused on the crop. I would suggest to reformulate in the abstract and conclusion, such as not to mislead the readers, and refer to another study to assert "[...] good natural cooling mechanism while also increasing the conversion efficiency of the solar system" (as it is not a conclusion from this study).
2) The definition of heat stress is still not very clear to me, an equation may be appropriate in this case. The uniformity of units (heat stress in % in Fig8 (which I understand better), and in Celsius in Fig9) may also be necessary for clarity.
Author Response
As attached pdf file

Reviewer 2 Report
The authors attempted to model the temperature and heat stress under the “Agrivoltaric” conditions. The contribution of this research is unclear, the manuscript is structured atrociously and should be completely restructured. Therefore, my recommendation for this manuscript is rejection.
The Abstract should be restructured so it follows some logical path, as it currently seems cluttered and not interesting. Currently, it is far too long (also disregards Instructions for Authors of this Journal) and must be more concise. Also, it is very important to distinguish the results of your research and from existing literature in the abstract, so that readers can evaluate your contribution to existing knowledge. The Introduction is far too long and becomes uninteresting fast. You should make it shorter and more concise, so it contains only the most valuable information for the reader without redundant data. This section seems like a part of the review paper and not of an article. A section containing Material and Methods should be better explained. Here must be written everything about how exactly did you collected results and analyzed them. Results and Discussion must be restructured to follow some logical path (which should be given in Material and Methods). All tables should be formatted equally, as it seriously affects the ability to review them. An English language check is recommended.
Specific comments:
Line 32: Around 32.9%? This seems like a definite value.
Line 56: “player”? Please avoid using this word.
Line 61: You already introduced this abbreviation.
Line 64: You introduced this abbreviation for the third time in 64 lines??
Table 1. This table is not necessary and you should explain these four studies in the text.
Lines 149-159: Please standardize the font type to make the reviewing process easier.
Lines 164, 182, 195, 207: You introduced or misplaced “RH” four times here. Did no one of eight authors read this manuscript thoroughly?
Lines 173-178: Should this be a part of the Introduction or Material and Methods?
Lines 188-200: This definitely belongs to the Material and Methods.
Line 251: Where is part a) of Figure 5? If it is there, it should be much more visible.
Lines 288-290: This belongs to the Introduction.
Lines 391-404 and 409-415: This represents a clear disrespect for the Journal and the reviewers. How can you possibly send something like this in a Q1 Journal?
Lines 417-418, 424-425: This just confirms what I said in the previous point.
Author Response
As attached pdf file

Round 2
Reviewer 2 Report
This version of the manuscript was improved after revisions. However, my impression of this manuscript is still that it can be structured better fundamentally. You made some improvements in the Abstract, but for about 80% of it, you wrote about the theoretical background of your work. Your contribution should be explained much more, as it was mentioned only briefly in the Abstract. Changes made in the Introduction produced little correction according to my previous comment. Materials and Methods, as well as Results and Discussion sections, are improved in contrast to the previous version. The quantity of figures in the manuscript does not justify the amount of useful information it contains and it should be reconsidered.
Author Response
As attached doc

This manuscript is a resubmission of an earlier submission. The following is a list of the peer review reports and author responses from that submission.
Round 1
Reviewer 1 Report
Review: Modeling of Stochastic Temperature and Heat Stress directly underneath Agrivoltaic conditions with Orthosiphon Stamineus Crops Cultivation
Othman et al.
Manuscript:
The manuscript by Othman et al. shows how the micro climatic condition below an Agrivoltaic system affects heat stress and temperatures of plants growing below the system. I think the paper includes a lot of great data and has the potential to make an impact in their field. However, I think to do so requires an expansion of either the Discussion/Results or the Conclusions, to really highlight the main findings. Also, an improved methods section with some key details about the design of the system will further strengthen the authors' work. The paper is very well written. The only grammatical areas where I noticed some issues were in matching singular and plural nouns with the correct verb conjugations. I think the paper can be greatly improved by putting greater detail in the figures and tables. I included a few specifics down below.
General comments:
On page 5 there are two nice paragraphs about VPD and the effects that rising VPD can have on different plant attributes. The authors do a nice job explaining some of the positive effects that VPD can have on plant growth, but I think it is important to expand the part about the potential negative effects. Increased transpiration can possibly lead to water stress. Some plants also have a tendency to close or partially close stomata when VPD reaches some threshold, and this would both reduce transpiration and growth.
Figure 3 is important, but it is difficult to read. Would it be possible to improve the quality so the words on the image are clearer? Also, a more detailed caption would help the reader understand the components a bit better.
Figure 6 is very neat. I think that the caption can be more informative to help highlight the importance of the figure. The three different panels should be explained. Are they the same location just taken at different hours? The scale for the temperature is a bit difficult to read, perhaps it would be better placed to the side of the figure with a white background? I also think a scale would be useful to have an idea of the height o the image.
Figure 7 is very important and informative. I think that the figure caption requires greater explanation. The legend and colors should be explained. The units of temperature as well as what the numbers on the x-axis refer to should also be clarified.
Table 2. I think that either the caption or a footnote needs to be added to explain what the units are and what the Tg, T1ft, etc mean. Also, why is the maximum value for VPD 0.548 and the minimum 2.005?
Table 3. How many different sample points were there? I may have missed this, but I am not sure I have found it in the methods. Over how many days were the temperatures measured? Like the other tables, I think the temperature abbreviations need to be explained somewhere in the caption or a footnote of the table.
Figure 8. How exactly was heat stress measured?
Table 4. What does it mean by the percentage of heat stress occurrence? Can this be expanded on a bit in the methods?
I think the Results/Discussion section can be expanded to include a more detailed interpretation of the results (which are really interesting and important for this field of study). There is a great opportunity here to put this work in the context of work elsewhere – at both a local and global scale. I think the Conclusion seems to include a lot of results and very little discussion. One potential choice would be to expand the Conclusions instead to put the work in a broader context and pull key findings from the Results/Discussion to highlight here.
Line comments:
It is a bit tricky to give precise line comments because the lines aren’t numbered, but I will do my best here.
Page 2 last paragraph: should “Just about all solar PV model in the market be…” be “Just about all solar PV models in the market be…”
Page 3 second paragraph: should “…and is a part of introducing modern aspect to community agriculture [13].” Be “…and is part of introducing modern aspects to community agriculture [13].”
Page 4 first bullet point: should “The existing solar PV farms does not…” be “The existing solar PV farms do not…”
Page 6. What is the average height of the solar arrays? And is the positioning of the sensor that records wind speed and RH unobstructed from the solar arrays and other vegetation?
Reviewer 2 Report
The manuscript entitled "Model of Stochastic temperature and heat stress directly underneath agrivoltaic conditions with orthosiphon stamineus crops cultivation" by Othman et al intents to present the temperature profile and heat stress conditions for crops under a PV system. The manuscript is quite difficult to follow, probably because no clear objective is stated from the starts, "This paper shares some new information [...]" being too vague. The introduction is not reaching the point of technicity shown later in the analysis, and therefore does not point out the specific questions that have to be answered. The definition of the heat stress is not clearly presented, and beyond is applied in conditions not specific to the plant cultivated (orthosiphon, while some comparison is done with tomato results). For the set-up, the distribution of placement of the PVs is probably crucial on the results, as well as the way the plants get watered and stage of growth, the two latter not being described. All of that would need a discussion, so as to be transferable. The analysis of 5 days only may be too short depending on the question to be answered, but may be sufficient if the study would be well framed, with an extensive description of the weather (including insulation conditions). As I did not find the definition of heat stress, I cannot be sure as well that the units of heat stress is represented on figure 9.
As a conclusion, an experimental set-up has served to collect data of temperature at different heights that could be useful, some interesting conclusions could emerge (Figure 8 (no label or caption)), but the study presented in the manuscript definitely lacks of a definite frame, with clear research questions, subsequent to other studies (literature).